# Are the Dental Guidelines for Early Dental Visits and Fluoridation Measures Supported by Pediatricians, and What Are Their Caries Prevention Efforts?

**DOI:** 10.3390/jcm11051159

**Published:** 2022-02-22

**Authors:** Antje Geiken, Louise Holtmann, Christian H. Splieth, Jonas Conrad, Christof E. Doerfer, Christian Graetz

**Affiliations:** 1Clinic for Conservative Dentistry and Periodontology, University Schleswig-Holstein, 24105 Kiel, Germany; holtmann@konspar.uni-kiel.de (L.H.); conrad@konspar.uni-kiel.de (J.C.); doerfer@konspar.uni-kiel.de (C.E.D.); graetz@konspar.uni-kiel.de (C.G.); 2Department of Preventive and Pediatric Dentistry, University Greifswald, 17475 Greifswald, Germany; splieth@uni-greifswald.de

**Keywords:** early childhood caries, primary teeth, fluoride, caries prevention, pediatrician

## Abstract

(1) Background: In Germany, new recommendations for dental examinations of children and the use of fluorides have been introduced. The pediatrician (PA) should refer the patient to the dentist for dental examinations and check-ups (DEs) from the sixth month of age. Therefore, our aim was to determine with a questionnaire the extent to which PAs find DE useful, make referrals for DE and recommend fluoride. (2) Methods: The nationwide empirical survey was conducted with a self-developed and validated standardized online questionnaire. In addition to personal information, 16 items were collected. Agreement with the items was recorded using Likert scales. The data were primarily analysed descriptively. (3) Results: 696 PAs participated in the survey (age: 51.7 (8.4) years, women/men: 428/286 (61.5/38.5%). A total of 11% of PAs found referral by eruption of first tooth very important (important/neutral/unimportant: 13.8/32/43.2%), compared to 70% for complete deciduous teeth (21.3/7.3/1.4%). A total of 48.8% of PAs always recommended fluoridated toothpaste from the first tooth (often/occasionally/rarely/never: 18.3/7.8/8/17.1%) and 50.6% completely refused to recommend fluoride-free toothpaste (always/often/occasionally/rarely: 9.8/9/14.7/15.9%). A total of 44.8% never recommended the use of fluoridated toothpaste if the child cannot yet spit (always/often/occasionally/rarely: 19.2/13.9/7.8/14.3%). (4) Conclusions: Among PAs, referral to DEs was increasingly implemented as children grew older. Specific fluoride recommendations were accepted.

## 1. Introduction

Early childhood caries (ECC) affects more than 600 million children worldwide. It is the most common preventable disease [1], shows rapid and aggressive progression, and can have various consequences in its course [2]. In the oral cavity, there is an immediate risk of toothache as an acute or chronic event. In addition, reduced food ingestion, abscess formation, damage to dental buds of permanent dentition and masticatory dysfunction are possible. Development can also be disturbed. The children can suffer from sleep disorders, feelings of inferiority, or deficiencies in language formation, such as sigmatism, and show a reduced quality of life compared to their healthy peers [3,4,5]. A child’s first dental contact has a major impact on their later attitude towards oral health and adherence to dental treatment [6,7]. A late presentation of children with a high caries risk in a dental practice often means that this group of patients has already developed carious lesions to be treated. Due to the young age and the resulting lack of adherence of the children, regular outpatient dental treatment is often not possible. Instead, complex rehabilitation under intubation anaesthesia must be carried out, which is associated with general medical risks for the child and higher costs for the health system [8].

An early visit to the dentist can help prevent the development of ECC. New guidelines for dental visits for children have been in effect in Germany since 2019. Previously, the first visit to the dentist was scheduled between 30 and 72 months of age, but this should be considered too late for children at risk of ECC, as carious defects may have already developed. The new guideline recommends the first visit at the first deciduous tooth (DE 1a) (at the age of 6 months).

Further examinations follow at the age of 10 to 20 months (DE 1b) and from 21 to 33 months (DE 1c).

Pediatricians see most children regularly in the first years of life and should carry out caries prevention measures as part of the early detection pediatric examinations (PEs). There are a total of nine early detection pediatric examinations (PE 1-9) from birth to the age of 6.

For example, at PE 5, they should advise on nutrition and breastfeeding, caries prophylaxis with fluoride, oral hygiene and a tooth-friendly diet, and refer children for dental examination and check-up (DE). PE 6 and PE 7 additionally include an examination of the teeth and mucous membranes for abnormalities and explicit advice on oral hygiene (dental care) and a tooth-friendly diet. They should also make parents aware of dental examinations and check-ups, and they therefore have a special responsibility in this age group [9,10] (Table 1).

In addition to early preventive and control visits, fluorides are one of the most important pillars in caries prophylaxis. With regular use of fluoridated toothpaste, a caries-inhibiting effect can be observed, so fluorides are considered the main reason for the decrease in caries in children and adolescents [11,12]. However, until 2021, the recommendations of pediatricians differed from those of dentists. The German Society for Pediatrics and Adolescent Medicine (DGKJ) and the German Academy for Pediatrics and Adolescent Medicine e. V. (DAKJ) have advised against fluoridated toothpaste due to concerns about swallowing and have recommended the administration of fluoride tablets from birth. These inconsistent guidelines have historically led to confusion among parents about the use of fluoride. Thus, a consensus became mandatory, and since 2021, there are now uniform fluoride recommendations from both specialists, which should simplify and standardize the recommendation [13].

However, for daily routine, little knowledge is available regarding the interaction between dentists and pediatricians. Therefore, the aim of this study was to investigate which fluoride recommendation and at what age German pediatricians recommend the first visit to the dentist.

## 2. Materials and Methods

### 2.1. Study Population

The current study was conducted among pediatricians in Germany by means of an online survey (observation time: 31 January 2020 until 7 May 2020).

The study was conducted according to the current guidelines of the Declaration of Helsinki from 2013 (Fortaleza, Brazil), and the ethics committee of the University of Kiel has approved this study (AZ: D452/18). Informed consent was obtained from all subjects involved in the study.

A cover letter, a survey link and a QR code were sent via the German professional association of pediatricians (BVKJ) office to 5406 pediatricians working throughout Germany who are registered in the BVKJ email register. The cover letter and the introductory text of the survey informed the subjects about the content of the questionnaire, the research question and the voluntary anonymous participation. Participants were not promised any benefits or financial incentives for their participation.

Reminders were sent by e-mail via the BVKJ after four and eight weeks following the initial announcement of the study survey in order to increase the response rate. Directly addressing non-responders to increase participation motivation was not possible due to the anonymous questionnaire design. Participants who are not in private practice but, for example, work in the university/clinical sector or in public service, students and pediatricians who are no longer active or retired were excluded. Questionnaires that were abandoned prematurely, i.e., abandoned by participants after the introduction page or abandoned after age, gender, and place of practice (state), were not completed, or incorrect questionnaires, i.e., no information on age, gender, occupation, or state, or incorrect information, such as age in xx or 00 years, were also not included.

### 2.2. Questionnaire

The questionnaire was tested by a focus group of 30 pediatricians from the BVKJ. Furthermore, the survey was validated by five experts in the fields of pediatric medicine, dentistry and web-based surveys at the University of Kiel and transformed into an online survey using a web-based survey tool (Unipark, QuestBack GmbH, Cologne, Germany). The questionnaire was divided into two sections. In the first part, the demographic information of the participants was collected, e.g., work in a private practice, age, gender and location of the practice (federal state).

The second part asked about the referral activity to a dentist and implementation of caries prophylaxis measures in children under 33 months (nine items) as well as the fluoride recommendations made (one item divided into seven sub-items). Likert scales were used to grade the answers for the given options, e.g., 1 to 5, with 1 representing “very rarely” and 5 representing “very often”.

It was not mandatory to answer all questions in the second part, so the total number per response could vary.

### 2.3. Statistical Analysis

There was no obligation to answer all questions. The graphical representations were created with Microsoft Excel 2011 for Mac (version 14.3.2, Microsoft Corporation, Redmond, WA, USA). Statistical analysis was conducted using SPSS for Mac, version 24 (SPSS Inc., version 24.0.0, IBM Corporation, New York, NY, USA). The significance level was set at 95% of statistical probability (*p* < 0.05). The analysis of the data was primarily descriptive, indicating the percentage frequencies, means, and Likert scales as medians with reference to the lower/upper quartile. The test for normal distribution with the help of a histogram and a box plot also revealed a curve that was not normally distributed.

## 3. Results

In total, the response rate was 23.4%. A total of 696 pediatricians (*n* = 428 (61.5%) women/*n* = 268 (38.5%) men) took part (Figure 1). The mean (SD) age of the included participants was 51.7 (8.4) years (Table 2).

Most of the subjects came from the federal state of North Rhine-Westphalia (*n* = 126/8.1%) (Table 3).

The pediatricians showed little referral activity to the dentist to clarify abnormalities on teeth and gingiva at the age of 6 to 24 months (9.9 (15.2)% (range 0–100)). The referral activity for early dental diagnosis examinations in children under 33 months of age was significantly higher (47.2 (42.2)% (range 0–100)).

The majority of the pediatricians (*n* = 313/45%) informed parents about the possibility of a dental screening examination, and 120 (17.2%) only provided information for parents who asked questions or seemed interested. A total of 65 (9.3%) of all participants did not inform the parents about dental screening examinations at all. Only a small number (*n* = 73/10.5%) of them expressed the opinion that there was insufficient time for referral to the dental practice. A total of 327 (47%) participants did not face any difficulties with a referral.

In a subgroup analysis of free text answers, 43 (35.5%) of 121 stated that dentists do not perform early dental examinations. Additionally, 25 (20.7%) indicated the young age of the children as a reason for not referring them to a dentist. Ten participants stated that there were not enough dentists (pediatric dentists) specializing in children.

Interestingly, the majority (*n* = 293/43.2%) felt a referral from the appearance of the first tooth was unimportant. Only 94 (13.8%) participants rated them as important and 75 (11.0%) as very important. Nearly double the number of participating pediatricians (*n* = 181/26.2%) considered a referral 6 to 24 months to be very important and 240 (34.8%) to be important, while only 72 (10.4%) pediatricians rated it as unimportant. With a similar trend as older age, referrals at full deciduous dentition were considered very important (*n* = 480/70.0%), whereby now only 10 (1.4%) participants considered a referral as unimportant (Table 4). The number of participants (679, 686, 690) varied for the different questions due to the fact that there was no obligation to answer the question.

### 3.1. Specific Scenarios of Caries Prevention for Children between 6 and 33 Months

All 696 participants validly answered the questions on ‘Caries prevention education’, ‘Education on oral hygiene measures’, and ‘Nutritional counselling in connection with caries prevention’. Four participants did not answer the question “Training parents in oral hygiene measures for their own child,” so 692 responses could be analysed. The majority of respondents always (*n* = 500/71.8%) and often (*n* = 127/24.7%) provided parents with caries prevention education. Similarly, 426 (61.2%) participating pediatricians always provided information to parents about oral hygiene measures for children. A lower number of 305 (43.8%) participants reported providing nutritional counselling to the parents of the children always, only 25 (3.6%) rarely and 10 (1.4%) not at all. A small number of 42 (6.1%) pediatricians always practised oral hygiene measures with the child’s parents, whereas the majority (*n* = 359/51.9%) did not do so at all (Table 5).

### 3.2. Fluoride Recommendations for Children between 6 and 33 Months

The question “Recommendation of fluoridated toothpaste from the first tooth” was answered by 690 participants, 683 answered “Recommendation of fluoridated toothpaste when the child can spit” and 680 answered “Recommendation of fluoride-free toothpaste”. Fortunately, nearly half of all pediatricians (*n* = 337/48.8%) always recommended fluoridated toothpaste for children; only 118 (17.1%) did not recommend it at all and completely rejected its use. A total of 19.2% (*n* = 131) of all participants always recommended the use of a children’s toothpaste only once the child was able to spit out, whereas the majority (*n* = 306/44.8%) did not recommend this. Contrarily, the majority (*n* = 344/50.6%) always decline to recommend fluoride-free toothpaste, while only 67 (9.8%) pediatricians recommend it (Table 6).

## 4. Discussion

### 4.1. Caries Prevention and Oral Hygiene

The majority of the study participants seem to be aware of their responsibility as the first medical contact for parents and also take caries-preventive measures. According to their statements, the study participants always carry out caries education (71.8%), oral hygiene measures (61.2%) and caries-preventive nutrition counselling (43.8%). These results prove that a majority of pediatricians take the preventive task of caries education seriously and comply with it. In contrast, 51.9% of them do not conduct any oral hygiene training for parents at all, while only 6.1% always do so. Yet, it is precisely the training of parents in good oral hygiene that would be an important point for the child, because education alone does not seem to be sufficient [14].

The spatial conditions in a pediatric practice seem to make the training of oral hygiene measures difficult, whereas dentists could easily take over these tasks and thus relieve the pediatric colleagues. However, this also requires children to be regularly and consistently referred to dentists by pediatricians.

According to the international recommendations [15,16], the German legislator also responded with new early dental examinations and permitted the first visit at the age of 6 months and within the limitations of the current online survey. The pediatricians refer up to 47.2 (42.2)% of their patients to a dentist in an average month (mean (SD)). However, many of the participating pediatricians did not appreciate the value of an early visit at the first deciduous tooth, one of the signs that early dental examinations and check-ups (DE 1a–c) were assessed very differently. According to our study results, the majority of the participating pediatricians ascribed no importance to a referral at the time of the first tooth. However, they felt that a referral parallel to PE 5-PE 7 was important, and even very important when the deciduous dentition was complete.

Wagner and Heinrich-Weltzien [17] asked Thuringian pediatricians about recommendations for the first visit to the dentist. The cohort was 43.9 (9.2) years old and 76.7% female. The choices were referral from the first tooth or referral after the first, second or third or fourth birthday. A total of 63% of the pediatricians recommended a referral after the second birthday and only 9% after the first tooth. Ismail et al. [18] surveyed 493 pediatricians in the USA on whether they followed the American Academy of Pediatric Dentristry (AAPD) recommendations for a dental visit from the first birthday. Interestingly, 63% of these American pediatricians recommended a visit from the age of three for children with a low caries risk, and 91.5% recommended the earliest possible presentation in cases of an increased caries risk. However, the question of to what extent pediatricians are able to correctly assess the risk of early childhood caries and actually implement diagnostics for signs of initial caries remains unanswered.

In contrast, a Brazilian group of pediatricians and doctors in practice, with a mean age of 36.7 years [range 23–70 years], assessed screening as useful from the first birthday, or even earlier [19]. A Europe-wide study by Hadjipanayis et al. [20] supported the recommendations of the above studies. The majority (43%) of these European pediatricians (age 53 (10) years, 58% female) recommended, without the control of caries, the first visit to the dentist from the age of three years, while only 7% recommended it for children aged under one year. The data from this survey confirm both our study situation and the overall prevailing picture. Although pediatricians and adolescent doctors are basically informed about the development of caries and the necessity of a dental presentation, they do not yet show sufficient awareness of the early development of the clinical picture of ECC, which is why referrals have so far been inadequate.

In the current investigation, the majority of the study subjects seem to be aware of their responsibility as the parents’ first medical contact and also take caries prevention measures. Our results prove that a majority of pediatricians take the preventive task of caries prevention education seriously and comply with it. In contrast to this well-performed preventive approach, in the current investigation, 51.9% of them do not implement oral hygiene training for parents at all, compared to only 6.1% who always do. However, it is precisely the training of parents in good oral hygiene for the child that would be an important point, as education alone seems insufficient [14]. Apart from the positive findings on referral by the participants of the current study, no statement can be made on the quality of pediatric education neither in the context of caries prevention nor for the instruction and motivation of oral hygiene at home. Other similar studies found that motivation for education among pediatricians is high, but deficits exist in knowledge about ECC disease [21,22,23,24,25]. Studies to date on pediatricians’ knowledge of how to recognise carious or even initial carious lesions in patients are limited. Oftentimes, data were obtained through self-reporting by participating pediatricians using questionnaires. In studies investigated with more reliable methods, e.g., a comparison of the ability of pediatricians to diagnose carious defects with pediatric dentists, the results showed weaknesses in both initial caries and cavities and a higher rate of false positives for the pediatrician group [26]. Unfortunately, this did not result in an increased referral activity for a dental consultation among the same pediatricians in this study, despite the caries they suspected. Yet, some other authors have found that pediatricians showed acceptable theoretical knowledge, but they are not sufficiently successful in practical implementation and pediatricians would like further training in this area [27].

Prakash et al. [28] surveyed 1044 pediatricians and general practitioners about their oral preventive interventions. Only 17.9% of pediatricians and 22.3% of general practitioners reported ever receiving training on the dental oral health aspect of their pediatric training.

### 4.2. Fluoride Recommendations

The current national consensus recommendations on fluoridation recommend the use of tablet fluoridation in combination with vitamin D prophylaxis until the first tooth [13]. From the first tooth until 12 months of age, a fluoride-free toothpaste and a combination of fluoride and vitamin D in tablet form can be given, or brushing with a fluoride toothpaste and a vitamin D tablet can be given. From the second year of life, only the use of a fluoride toothpaste is recommended. This has to be seen as a great challenge for caries prevention, as the early use of a fluoride toothpaste has historically been controversially discussed between dentists and pediatricians [29]. The latter recommend it when the child is undoubtedly able to control spitting out [29], which is oftentimes the case from the age of four. However, further efforts have to be undertaken to improve the knowledge of oral diseases of enamel and fluoride administration among pediatricians, as, e.g., in a regional survey in the year 2014, nearly 21% of the participants recommended fluorettes and brushing with fluoride-free toothpaste, but 45.9% also recommended fluorettes and fluoridated toothpaste, which greatly increases the risk of fluorosis. Brushing teeth from the first tooth was recommended by 35%, while 52% recommended brushing teeth from the second birthday [17].

The results of this recent survey indicate a change in trend: pediatricians always recommended fluoride toothpaste from the first tooth in almost half of the responses, and a majority of them did not recommend waiting until the child can spit before brushing with a fluoridated toothpaste. Fortunately, only a minority recommended fluoride-free toothpaste at all. This clearly contradicts the earlier studies from Germany mentioned above [17,30,31]. However, it should be noted that in this survey, the use of fluorettes could not be indicated directly, but only indirectly via the rejection of the recommendation of fluoridated toothpaste.

The results must also be seen in the particular context of the German national fluoride recommendations. These represent joint consensus recommendations by pediatricians and dentists and were published in 2021 [13]. According to the German recommendations, combination preparations of fluoride and vitamin D can be recommended. This is in contrast to the European and international fluoride recommendations, which recommend local fluoridation with toothpaste and attach less importance to fluoride supplements [32,33].

### 4.3. Limitations of the Online Survey

The external validity of our study results is limited and should be interpreted with caution. In terms of representativeness, the study does not correspond to a classical method in which a random sample is generated as a study cohort with the help of an existing directory.

The study participants were recruited from a group of pediatricians from the e-mail distribution list of the BVKJ member directory. This directory contains the data of 5046 pediatricians. Due to the pre-selection and non-specific recruitment of the subjects, a selection bias is present.

The 696 participants who responded to the call and completed the questionnaire in full could have a particular interest in the field of fluoridation and early dental examination and be more motivated than the national average of pediatricians in practice. A resulting distortion of the behaviour and opinion results cannot be ruled out.

## 5. Conclusions

The referral activity to new early dental examinations by directly addressing pediatricians offers the possibility to motivate more parents to undertake an early presentation to a dentist. Within the framework of the study and the existing restrictions, it could be shown that pediatricians carry out caries preventive measures. However, the majority of the participating pediatricians consider the newly introduced possibility of referring children to the dentist and DE earlier to be more important the older the children are.

Therefore, efforts must be intensified to increase pediatricians’ willingness to make early referrals. We feel that structured training could help to sensitize pediatricians to the clinical picture of ECC and thus make clear the purpose of an early referral to the dentist. From a dental perspective, the results about the fluoride recommendations are very pleasing. The majority of respondents recommend fluoridated toothpaste, which we interpreted as renewed consideration of the pediatricians in recent years.

## Figures and Tables

**Figure 1 jcm-11-01159-f001:**
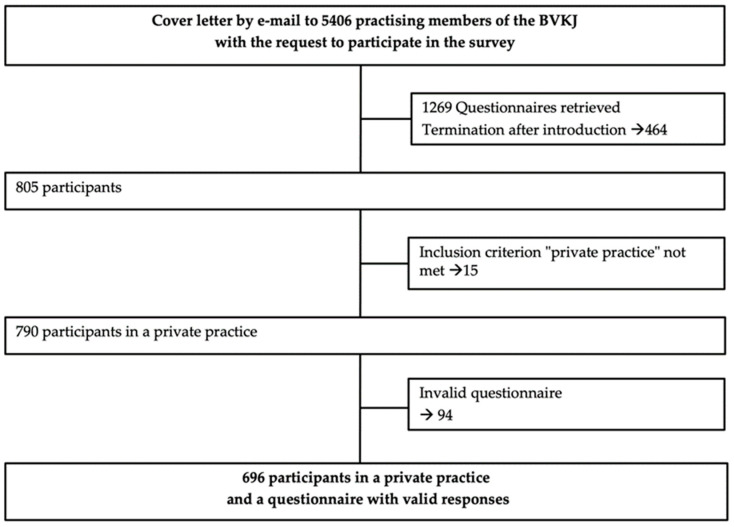
Recruitment scheme of the participating padiatricians in private practice.

**Table 1 jcm-11-01159-t001:** Current guidelines in the age group of 6–33 months. Referrals to the dentist by the pediatrician and early dental examination and check-up.

Early Detection Pediatric Examinations (PE)	Referral to a Dentist	Dental Examination and Check-Up (DE)
PE 5 (6–7 months)	Clarification of abnormalities of teeth and mucous membranes	DE 1a DE Pr * (6–9 months)
PE 6 (10–12 months)	Clarification of abnormalities of teeth and mucous membranes	DE 1b DE Pr * (10–20 months)
PE 7 (21–24 months)	Clarification of abnormalities of teeth and mucous membranes	DE 1c DE Pr * (21–33 months)

* DE Pr Practical oral hygiene instructions to the caregivers.

**Table 2 jcm-11-01159-t002:** Demographic data of the participants.

Demographic Data	Result
Number of participants	696
Female/Male (%)	428/268 (61.5%/38.5%)
Age (years)(SD) (range)	51.7(8.4) (31–76)
Female: Age (years)(SD) (range)	50.8(8.5) (31–70)
Male: Age (years)(SD) (range)	53.2(8.1) (35–76)

**Table 3 jcm-11-01159-t003:** Number of participants by federal state (in alphabetical order) in absolute values and in percent (%).

Federal State	Number of Participants
Baden-Wuerttemberg	107 (15.4%)
Bavaria	101 (14.5%)
Berlin	25 (3.6%)
Brandenburg	21 (3.0%)
Bremen	12 (1.7%)
Hamburg	21 (3.0%)
Hesse	49 (7.0%)
Mecklenburg-Western Pomerania	11 (1.6%)
North Rhine-Westphalia	126 (18.1%)
Lower Saxony	65 (9.4%)
Rhineland-Palatinate	25 (3.6%)
Saarland	10 (1.4%)
Saxony	38 (5.5%)
Saxony-Anhalt	19 (2.7%)
Schleswig-Holstein	40 (5.7%)
Thuringia	26 (3.8%)

**Table 4 jcm-11-01159-t004:** Perceived significance of early dental visit by age group in absolute values and in percent (%).

	Total	Very Important	Important	Neutral	Unimportant
From the first tooth	679	75 (11.0%)	94 (13.8%)	217 (32.0%)	293 (43.2%)
Parallel to the DE 5-DE 7 (6–24 months)	690	181 (26.2%)	240 (34.8%)	197 (28.6%)	72 (10.4%)
From the complete first dentition	686	480 (70.0%)	146 (21.3%)	50 (7.3%)	10 (1.4%)

**Table 5 jcm-11-01159-t005:** Measures carried out in children between 6 and 33 months as part of caries prevention in absolute values and in percent (%).

	Total	Never	Rare	Occasionally	Often	Always
Caries prevention education	696	1 (0.1%)	5 (0.7%)	18 (2.6%)	172 (24.7%)	500 (71.8%)
Education on oral hygiene measures	696	2 (0.3%)	9 (1.3%)	42 (6.0%)	217 (31.2%)	426 (61.2%)
Nutritional counselling	696	10 (1.4%)	25 (3.6%)	96 (13.8%)	260 (37.4%)	305 (43.8%)
Oral hygiene training	692	359 (51.9%)	160 (23.1%)	90 (13.0%)	41 (5.9%)	42 (6.1%)

**Table 6 jcm-11-01159-t006:** Fluoride recommendations of pediatricians in absolute values and in percent (%).

	Total	Never	Rare	Occasionally	Often	Always
Fluoridated toothpaste from the first tooth	690	118 (17.1%)	55 (8.0%)	54 (7.8%)	126 (18.3%)	337 (48.8%)
Fluoridated toothpaste, only if spitting out is possible	683	306 (44.8%)	98 (14.3%)	53 (7.8%)	95 (13.9%)	131 (19.2%)
Fluoride free toothpaste	680	344 (50.6%)	108 (15.9%)	100 (14.7%)	61 (9.0%)	67 (9.8%)

## Data Availability

Not applicable.

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
