# Peer review of "Are the Dental Guidelines for Early Dental Visits and Fluoridation Measures Supported by Pediatricians, and What Are Their Caries Prevention Efforts?"

_jcm, 2022, doi:10.3390/jcm11051159_

Round 1

Reviewer 1 Report

General comments:

Table 2 is missing or the tables are wrongly numbered.

The questionnaire or the questions and answer options should be attached to make the method section and the results easier to understand.

Demographic data was asked for, but it was not analyzed or presented in full.

The methods section should be more detailed.

How many participants were excluded according to the stated criteria and for what respective reason? (Diagram would be helpful)

There is still unused potential in the data. Are there differences in the data from older and younger pediatricians? Is there a difference between practices in rural areas (without a dentist or pediatric dentist nearby) and metropolitan areas? Are there differences between East and West Germany (there were specialized pediatric dentists in the former GDR)?

Title:

  • I think the title is too general and thus somewhat misleading. - Rather, it is an article about the question: Are the dental guidelines for early dental visits and fluoridation measures supported or considered important by paediatricians and what are their caries prevention efforts?

Abstract:

  • There is no indication in the aim of the study that the fluoride recommendations were asked.
  • Line 17: Sentence starts with "one" percent. But that should be 11 %.
  • Line 20: wrong percentage of 14 %
  • The conclusion contradicts a little the conclusion section in the text. This leads to confusion.

“the recommendations are considered” vs. “the overwhelming majority perceived it as unimportant”

Introduction:

  • Table 1: **FLA is not included in the table and therefore does not need to be explained.
  • Lines 71 - 73: The question about the paediatricians' own measures for caries prevention is missing.

Material and Methods:

  • Line 82: Abbreviation BVKJ must be explained
  • Lines 89, 90: What exactly does premature termination mean? Is that different from incomplete? What exactly could be errors? Please explain in detail.

Results:

  • Table for demography missing.
  • Line 117 and 119: what does the second percentage say? (15.2%, 42.2%)
  • Line 122: Do I understand it correctly that the paediatricians do not inform about the dental examinations, even if the parents would ask for it, for example?
  • Line 125: Percentage is missing
  • Table 3: Please explain the missing values and check the percentages
  • Line 143: “…did not give valid answers…” Please explain in the method section how the wrong answers were possible. This is difficult to understand in multiple-choice questions.
  • Lines 150 and 151: Please check the percentages in the brackets
  • Table 4: What is the definition of occasional or where are the limits?; Wouldn't it perhaps be better to combine the groups for better clarity (never and rare, occasionally and often, always)?
  • Table 4: Please check the percentages.
  • Table 4: The sentence under the table must be deleted.
  • Lines 159-161: One of the two sentences is redundant.
  • Line 162: “One hundred thirty-one…” This is the only number written as a word. I think it should also be written as a number. (not at the beginning of the sentence)
  • Line 164: Please check the Percentage
  • Table 5: Please explain the missings.

Discussion:

  • Line 172 last word: what does this mean “monthly”?
  • Line 173: What is the 42.2%?
  • Line 179: Do pediatricians always check the exact number of teeth? Was there a question about this?
  • Line 180: The number of participants in the study mentioned should be included, as it is relevant and was also reported in the other studies below.
  • Line 201 and 202: This is a very important aspect, but it was not adressed with in this study and, as far as I can see, was not criticized by the pediatricians. It should be deleted.
  • Line 209 and 211: Please check the percentages.
  • Lines 223-226: Pediatricians recognize the problems but do nothing. What is the explanation or assumption of the authors of this study why this is so?
  • Lines 248-250: Please explain the statement in more detail.

Conclusion:

  • Lines 273 and 274: This sounds very negative and in my opinion contradicts the statement in the abstract.

References:

  • Lines 315, 339, 360: Please correct

Reviewer 2 Report

There are several grammar mistakes to the manuscript. Also, I strongly disagree with the fluoride recommendations paragraph, Vit D and fluoride tablets??? Maybe the national recommendations should be revised. 

Round 2

Reviewer 2 Report

No comment, very nice revised manuscript